# Generic Characterization of Electrical Test Benches for AC- and HVDC-Connected Wind Power Plants

Behnam Nouri[1], Ömer Göksu[1], Vahan Gevorgian[2], and Poul Ejnar Sørensen[1]

[1]Department of DTU Wind Energy, Technical University of Denmark, 4000 Roskilde, Denmark
[2]National Renewable Energy Laboratory, Golden, CO, USA
**Correspondence:** Behnam Nouri (beno@dtu.dk)

**Abstract.** The electrical test and assessment of wind turbines are going hand in hand with standards and network connection requirements. In this paper, the generic structure of advanced electrical test benches, including grid emulator or controllable grid interface, wind torque emulator and device under test, has been proposed to harmonize state-of-the-art test sites. Besides, the modern wind turbines have been featured by new capabilities, such as grid-forming, black start, harmonic rejection and frequency support, to increase the robustness and reliability of renewable-energy-based grids. Furthermore, the increasing challenges, such as harmonic resonances and grid interactions, are compromising wind energy integration into power systems. Therefore, it is necessary to develop new and revised test standards and regulations. This paper proposes a generic test structure within two main groups, including open-loop and closed-loop tests. The open-loop tests include the IEC 61400-21-1 standard tests as well as the additional proposed test options for the new capabilities of wind turbines, which replicate grid connection compliance tests using open-loop references for the grid emulator. Besides, the closed-loop tests evaluate the device under test as part of a virtual wind power plant and perform real-time simulations considering the grid dynamics. The closed-loop tests concern grid connection typologies consisting of AC and HVDC, as well as different electrical characteristics, including impedance, short circuit ratio, inertia, and background harmonics. The proposed tests can be implemented using the available advanced test benches by adjusting their control systems. The characteristics of a real power system can be emulated by a grid emulator coupled with real-time digital simulator systems through a high bandwidth power-hardware-in-the-loop interface.

## 1 Introduction

Wind energy has been one of the most promising renewable energy sources used worldwide, mostly located onshore. Besides, a better quality of the wind resource and larger suitable areas in the sea have made offshore installations a considerable choice for Wind Power Plants (WPPs). To date, the total installed capacity has reached 592 GW with 23 GW share of offshore in 2018. The new total installations would continue with more than 55 GW each year by 2023 (GWEC (2018); Wind Europe (2018)).

The increasing installed capacity of Variable Renewable Generation (VRG) has concerned power system operators in terms of stability and reliability of the overall power system. Consequently, new interconnection requirements, standards, and market mechanisms are evolving in various parts of the world for VRGs, including wind power, to provide various types of essential reliability services to the grid – the role that has been typically reserved for conventional generation (NERC, 2015). Further-

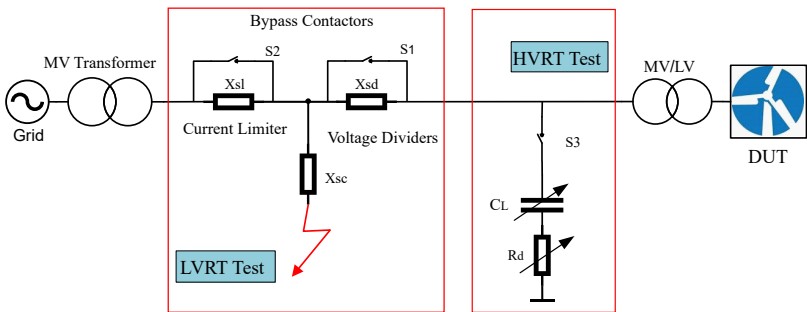

**Figure 1.** The basic structure of impedance-based topology for LVRT and HVRT capabilities tests (Ausin et al. (2008); Langstadtler et al. (2015)).

more, the industry has focused on collaborations and harmonization to achieve the technical and economic benefits of a uniform technology and market, especially in Europe (IRENA (2018); Sørensen et al. (2019)). In this way, for instance, the European Commission has regulated international requirements for AC- and HVDC-connected power-generating modules as well as HVDC systems (Commission Regulation 631 (2016); Commission Regulation 1447 (2016)). Consequently, compliance test standards are needed to ensure the power quality and performance of VRGs, especially WPPs.

Compliance test methods are in line with relevant network codes and standards. Furthermore, wind technology has been matured by research, development, and demonstrations in industrial test sites and laboratories. Figure 1 illustrates the basic compliance test equipment, which had been proposed for Low Voltage Ride-Through (LVRT) capability test in Ausin et al. (2008) and is addressed as an example in IEC 61400-21-1 (2019). Recently, this structure has been adapted for High Voltage Ride-Through (HVRT) capability test as well (Langstadtler et al., 2015). In this topology, the voltage divider impedances ($X_{sd}$ and $X_{sc}$) are used for the LVRT test of the device under test (DUT). Also, the parallel capacitors ($C_L$) in series with damping resistors ($Rd$) are used for the HVRT test. $X_{sl}$ is used to limit the effect of tests on the utility grid by limiting the current flow from the utility grid during the test. The test apparatus structure shown in Figure 1 had proven to be a useful tool in the early stages of grid integration research and criticizing of utility-scale wind power. However, it has certain fundamental limitations, such as dependence on a stronger point of interconnections, uncontrollable dynamic change of impedance during test, and inability to replicate any evolving grid characteristics.

Primarily, power quality and transient performance during faults have been essential aspects, which needed to be tested and verified. However, by increasing trends towards 100% VRG-based grids, the VRGs are required to be developed and featured by advanced capabilities to ensure the robustness and reliability of such grids. In this way, the state-of-the-art Wind Turbines (WTs) are under development to be upgraded to more advanced features such as grid-forming, black start, and frequency support capabilities. These new features would necessitate test and assessment standards in the near future (Langstadtler et al. (2015); Asmine et al. (2017); Gevorgian et al. (2016)). Besides, by increasing wind power installations, the requirements and appropriate test methods are required to study the rising challenges such as harmonic resonances and control interactions

of WPPs in connection to different types of AC and HVDC transmission systems according to Hertem et al. (2016), Zeni et al. (2016) and Buchhagen et al. (2015). Thus, it is essential to adapt or define new regulations, standards, and compliance test methods to analyse the developments and issues regarding wind energy. To date, several standards and recommendations such as IEC, IEEE, DNV GL, and CIGRE have been published for design, simulation, operation, and test of electrical aspects of WTs (IEC 61400-21-1 (2019); IEEE Std 1094-1991 (1991); DNVGL-ST-0076 (2015); CIGRE TB 766 (2019)). The IEC standards as the leading international standards for the test and assessment of wind turbines have been reviewed in this paper.

In this paper, the authors aim to extend the state-of-the-art developments in wind energy towards harmonized test methods and propose additional test options to the standard tests to extend the applications of advanced industrial test benches in terms of research and development studies. In part 2, grid connection compliance tests, including typical grid connection topologies, IEC standards, and electrical test levels, have been introduced. Part 3 overviews the state-of-the-art industrial test benches and illustrates the generic structure of converter-based test equipment. In part 4, the electrical characteristics of different grids to be emulated in a test site have been studied and proposed. Finally, part 5 proposes the generic structure of test options consisting of the recommended tests in IEC standard as well as proposed additional test options for open-loop tests as well as closed-loop tests for WTs and WPPs.

## 2 Grid Connection Compliance Tests

The integration of wind energy into the power system has been one of the main challenges for the development of WPPs. The wind power can be transmitted either through AC or HVDC transmission systems to the main AC grids. Besides, there is an increasing trend to develop WPPs in offshore areas because of the higher power capacity of offshore winds and limited onshore sites (Wind Europe (2018); (Cutululis, 2018); (Kalair, 2016)). According to the European Wind Energy Association (EWEA) (Pierria et al., 2017), potentially, the European offshore wind power can supply Europe seven times more than its demand. Figure 2 illustrates a typical structure for AC and HVDC connections of offshore WPPs. As shown in Figure 2-a, the AC-connected offshore WPP connects to the main onshore grid through high voltage submarine cables and transformers. The shunt inductors are required to dampen the possible over-voltage phenomena caused by the capacitive effect of the AC cables. The typical structure of an HVDC-connected offshore WPP is illustrated in Figure 2-b, which consists of HVDC transmission cables, transformers, AC/DC converters, and harmonic filters of the converters. HVDC connection has economic advantages for long distances, especially in case of offshore WPPs (Hertem et al. (2016); Cutululis (2018); Kalair (2016)). Hence, the recent interests in wind energy are focused on offshore WPPs, and HVDC systems are required due to distances from the main AC grids. The collector system voltages in AC- and HVDC-connected WPPs are typically 33 kV and 34.5 kV in Europe and the U.S., respectively. Recently, several 66 kV collector systems in offshore WPPs have been demonstrated. Therefore, 66 kV seems to be a general trend in collector system design in the offshore wind industry (Wiser et al., 2018).

The development process of grid connection requirements -or network codes- and compliance test methods occurred by the maturation of wind energy technology. Besides, the industry is currently interested in the technical and economic benefits of international collaborations (Wind Europe (2018); NERC (2015)). Therefore, harmonized regulations and standards are in

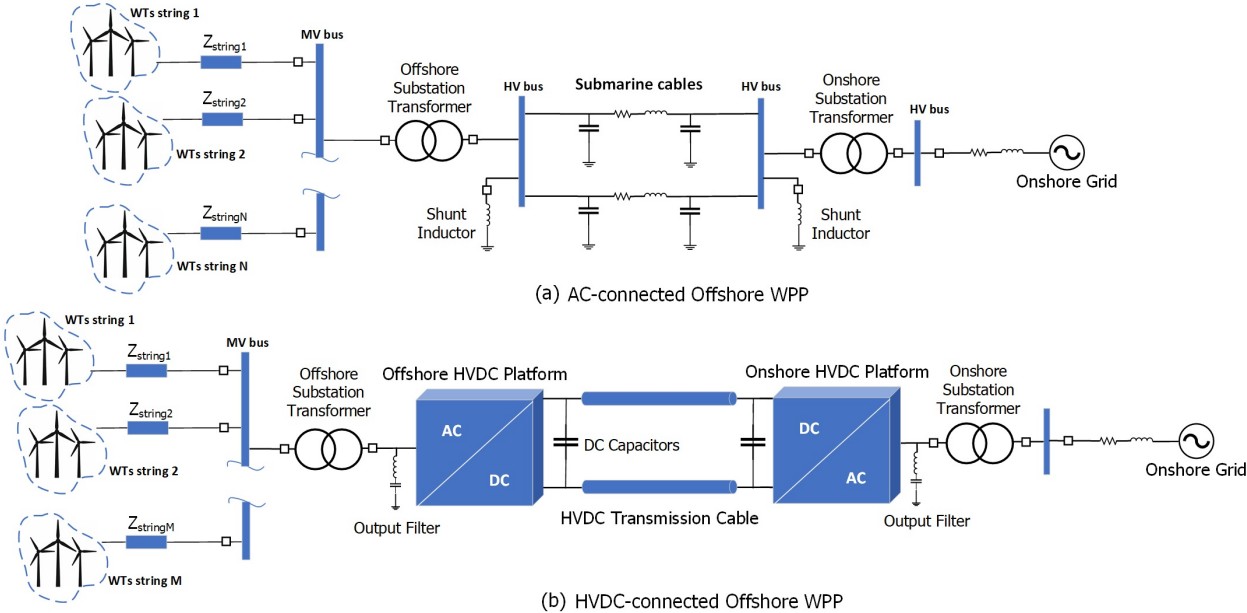

**Figure 2.** Typical structure of AC (a) and HVDC (b) connected offshore WPPs (Cutululis (2018); Kalair (2016)).

progress for design and performance assessment of WTs as well as WPPs. The development of European network codes and IEC standards are some of the best harmonization practices.

In European network codes, the requirements have been regulated for AC-connected offshore and onshore as well as HVDC-connected Power-Generating Modules (PGM) (Commission Regulation 631 (2016); Commission Regulation 1447 (2016)). According to (Nouri et al., 2019), the requirements for AC-connected offshore and onshore PGMs are mostly similar, while relatively different operation ranges and conditions have been considered for AC- and HVDC-connected PGMs. The AC and HVDC transmission systems impose different electrical characteristics on WPPs. Consequently, different control schemes and design considerations have been used for WTs and WPPs. Therefore in this paper, the authors propose to define a group of grid interaction tests considering the AC and HVDC connections for WTs using a converter-based test bench and emulate different grid characteristics for DUT. Although, reflecting all aspects of different grids is challenging, but at the same time, essential to assess the performance of WTs in a more realistic environment. Network code compliance tests and standards are critical factors in preserving the reliability and stability of WPPs. Thus, in the next section, IEC standards, as the leading international standards for test and assessment of wind turbines capabilities, have been reviewed.

## 2.1 IEC Standards for Assessment of Wind Energy

In 1988, Technical Committee 88 (TC88) of the IEC began its efforts to organize international standards for wind turbines as 61400 series. TC88 consists of several working groups, projects, and maintenance teams to develop and issue standards, technical reports, and specifications (Andresen et al., 2019). Initially, TC88 focused on power performance (i.e., power curve)

tests and structural and mechanical design. The works on electrical tests started in 1997 as IEC 61400-21 series by the working group WG21.

The second edition of IEC 61400-21 was published in 2008 to cover the definition and specifications for measurement and assessment of power quality characteristics for wind turbines. Currently, IEC TC88 WG21 is working on four new documents for the IEC 61400-21 series, where the title is changed from power quality characteristics to electrical characteristics appreciating that not only power quality characteristics are included (Andresen et al., 2019). To date, there is no IEC standard for testing the electrical characteristics of WPPs, but only for testing single WTs. Regarding the grid connection compliance assessment,

the evaluation of performance and quality of WPPs is based on measurements, simulations, and model validation tests (Ausin et al. (2008); Asmine et al. (2017); Andresen et al. (2019)).

    Recently, IEC 61400-21-1 is published and replaced the second edition of 61400-21. IEC 61400-21-1 specifies test methods for electrical characteristics of wind turbines (IEC 61400-21-1, 2019). Also, IEC 61400-21-2 specifies test methods for electrical characteristics of WPPs (Andresen et al., 2019). Concerning the growing issues regarding harmonics in WPPs, IEC

61400-21-3 aims to focus on harmonic modeling as a technical report. The IEC TR 61400-21-3 provides a starting point for the required frequency-domain modeling of wind turbines (IEC TR 61400-21-3, 2019). Furthermore, the IEC 61400-21-4 recommends a technical specification for component and subsystem tests (Andresen et al., 2019). IEC 61400-21-1 and 21-3 are published in 2019, while 61400-21-2 and -21-4 may be published in 2021.

    Besides, the IEC 61400-27 series specifies standard dynamic electrical simulation models for wind power generation. The

first edition of IEC 61400-27-1, published in 2015, specifies generic models and validation procedures for wind turbine models. Furthermore, the next edition is under development to expand the scope towards WPPs models in addition to the WTs models (Sørensen, 2019). The next edition consists of two parts: IEC 61400-27-1 specifying generic models for both WTs and WPPs, and 61400-27-2 specifying validation procedures.

### 2.1.1   Electrical test levels

According to the IEC-61400-21-1 (IEC 61400-21-1, 2019), the electrical characteristics to be simulated and validated for wind turbines consist of five different categories as power quality aspects, steady-state operation, control performance, transient performance or fault ride-through capability, and grid protection. The electrical characteristics of WTs can be measured and tested at different levels. The test levels consist of component test level (such as capacitors and switches), subsystem test level (such as nacelle and converter), field measurement at wind turbine level (or type test), and field test or measurement at WPP

level (IEC 61400-21-1, 2019). WT level tests can also be split into two subcategories: (a) testing of the full drive-train connected to low voltage test bench; (b) testing of the full drive-train connected to medium voltage test bench via WT's transformer with a full set of protection and switchgear (Koralewicz et al., 2017). The second option is closer to reality since it includes impacts of transformer impedance and configuration and protection settings on transient performance. In IEC 61400-21-1 (2019), an overview of the required and optional test levels for different test and measurement requirements is provided.

Nowadays, to have a flexible and economical solution for grid connection compliance tests and model validations, the trend is to perform tests at lower levels, such as WT and subsystem levels. The tests for WT and subsystem levels are mostly

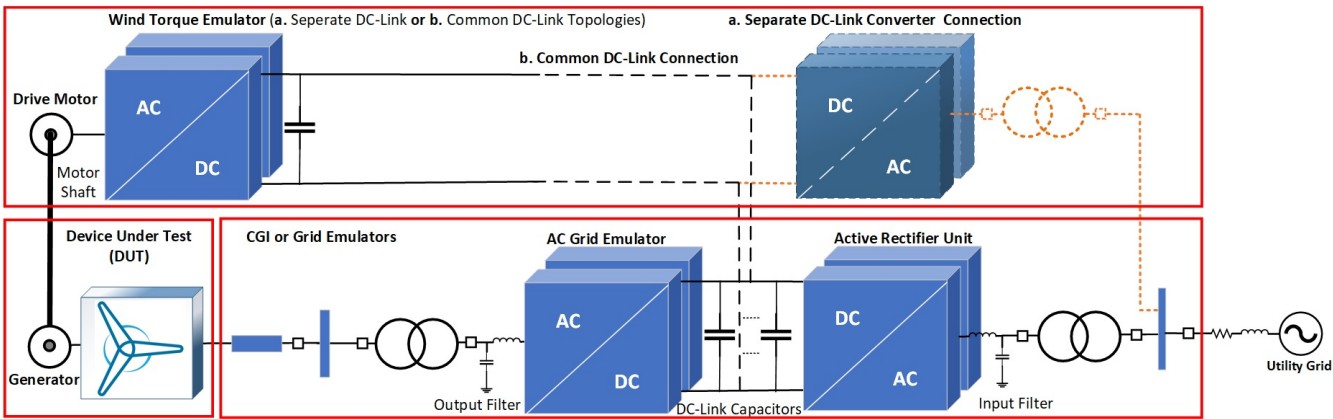

**Figure 3.** Proposed generic schematic diagram of a converter-based test bench.

implemented in a research and development environment or test sites. The test results concern wind farm developers and system operators in terms of WPP model validation and grid connection compliance, and WT manufacturers in terms of WT design and simulation model validations. This way, the results of tests are considered to be transferable and useful for the assessment of WTs as well as WPPs and developed simulation models (Ausin et al. (2008); Zeni et al. (2016); Koralewicz et al. (2017)). However, in some cases performing field tests and measurements are still necessary as reported in (Asmine et al., 2017). Accordingly, the Hydro-Québec TransÉnergie experience (Asmine et al., 2017) regarding the inertial response has shown that an adequate evaluation of the inertial response cannot be performed accurately at WT level and should include an evaluation performed at the WPP level. As another example, the power quality assessment of WPPs is either assessed using scaling rules of WT test results or accomplished by the assessment of online measurement data. The online monitoring is achieved during the first year of operation of the WPP (Asmine et al., 2017). The owners of power plants should secure that their connection to the local grid does not cause voltage distortion or fluctuation more than an acceptable range. However, the increasing challenges, such as harmonic resonances, grid interactions, and voltage and frequency stability issues, have proven the need for more extended analysis and assessment of WPPs. In this regard, the generic converter-based test bench and possible test and assessment solutions for WTs as well as WPPs are proposed in the next sections.

## 3  Generic Converter-Based Test Bench

Different electrical test benches as Controllable Grid Interfaces (CGI) have been reported for grid dynamics emulation in Ausin et al. (2008), Gevorgian et al. (2016), Espinoza et al. (2019), Espinoza et al. (2015), and Yang et al. (2012). The impedance-based test equipment in Figure 1 is only intended for the fault ride-through capability tests. A more advanced and flexible topology is a full-power converter-based CGI Yang et al. (2012), which is shown in Figure 3. This topology has been used in the latest industrial test benches and is studied in the next sections of this paper.

In MEGAVIND (2016), a mapping of global test and demonstration facilities serving the wind industry in Europe and the U.S. is presented by topics and locations. Accordingly, most of the latest industrial test benches are based on power electronic converters. The popularity of the converter-based test benches is because of enhanced control and test opportunities for both electrical and mechanical aspects of WTs. Converter-based test equipment provides emulation of unlimited test scenarios applicable to power systems of various sizes (sizeable interconnected power grids, island systems, or mini-grids) operating at both 50 Hz or 60 Hz, with full controllability over strength, imbalances and harmonic content of emulated grids. The generic schematic diagram of a converter-based test rig is shown in Figure 3. Generally, an industrial test bench for wind energy consists of three main parts: Device Under Test (DUT), wind torque emulator, and grid emulator or CGI. In Figure 3, the DUT is a WT nacelle. The CGI can also be used for testing of complete WTs in which case the wind torque emulator in Figure 3 is not required.

In Table 1, the specifications for some of remarkable advanced test sites are illustrated. As it is presented in Figure 3, the application of multilevel drive converter modules in parallel connections is a typical topology to establish a medium power and medium voltage source as grid and wind torque emulators (Averous et al. (2017); Gevorgian (2018); Jersch (2018); Rasmussen (2015) and Tuten (2016)). The multilevel converters, such as Three-Level Neutral Point Clamped (3L-NPC) and H-bridge topologies, are developed to achieve higher efficiency and lower harmonic distortion rather than conventional two-level converters and reduce the size of harmonic filtering and undesired interference.

According to the Table 1, a group of test benches such as available test setups in NREL (National Renewable Energy Laboratory, USA), Fraunhofer IWES (Fraunhofer Institute for Wind Energy Systems, Germany) and CENER (National Renewable Energy Centre, Spain), have used three-level NPC drive converters developed by ABB company. The ABB drive converters

**Table 1.** Comparison of different concepts applied in industrial test benches.

| Test Centre | CGI rating | Short circuit capacity | Torque Emulator rating | Converters Type | *N (ARU) | **M (AGE) | Converter Control | RTDS |
|---|---|---|---|---|---|---|---|---|
| LORC | 15 MVA | 30 MVA | 13 MW | 3-level NPC GE (IGBT) | 2 | 2 | AVC | no |
| Aachen | 3.5 MVA | 7.5 MVA | 4 MW | 3-level NPC GE (IGBT) | 1 | 1 | AVC | yes |
| NREL | 7 MVA | 40 MVA | 5 MW | 3-level NPC ABB (IGCT) | 1 | 4 | DTC | yes |
| F. IWES | 15 MVA | 44 MVA | 10 MW | 3-level NPC ABB (IGCT) | 2 | 3 | DTC | yes |
| CENER | 9 MVA | 18 MVA | 9 MW | 3-level NPC ABB (IGCT) | 1 | 2 | DTC | yes |
| Clemson | 15 MVA | 20 MVA | 7.5 and 15 MW | H-bridges TECO-Westinghouse (IGBT) | 2 | 2 | AVC | yes |

*N(ARU): Number of ARU modules, **M(AGE): Number of AC Grid Emulator modules.

are controlled by Direct Torque Control (DTC) method with Integrated Gate-Commutated Thyristor (IGCT) switches (ABB, 2018). On the other hand, in the second group, such as LORC (Lindø Offshore Renewables Center, Denmark) and Aachen (RWTH Aachen University, Germany), the converters are three-level NPC developed by GE company (General Electric). GE's medium power drive converters are controlled by Advanced Vector Control (AVC) using Insulated Gate Bipolar Transistor (IGBT) switches (GE, 2018). Besides, different types of converters would be utilized in a test site. For instance, the drivetrain test facility at Clemson University is established using multilevel H-bridge drive converters developed by TECO-Westinghouse company (Tuten, 2016). Each converter developer utilizes different components, control methods, and interface algorithms. However, all of the test benches should be able to perform tests according to the standards and research objectives, and minimize the effect of non-ideal emulation of a real test environment for DUT. In most of the test sites, a Real-Time Digital Simulation (RTDS) system is used to get to a dynamic online model of the grid as well as the overall system. The main limitation of converter topology shown in Figure 3, as well as any converter-based test rigs, is limited over-current capability. This constraint can be addressed by over-sizing the MVA rating of the test side converter similar to what was done in NREL's CGI (7 MVA continuous power rating but capable of operating at 40 MVA short circuit capacity during 2 seconds (Gevorgian, 2018) ) as given in Table 1. Over-sizing of converters for this purpose is costly but is necessary for LVRT testing of Doubly-Fed Induction Generator (DFIG) type WT, which can produce higher levels of short circuit current contribution.

The establishment of test equipment would be based on different criteria, objectives, and motivations. The majority of companies have plans to develop their sites as such to be able to test a wide range of WTs, including medium power to higher power ratings, that are mostly for offshore applications. According to GE (2018), the new trends in the development of grid simulators are as follow:

– Higher power ratings: up to 24MW rating and 80MVA short circuit power.

– Grid impedance emulation: virtual impedance emulation using the converters control system.

– Higher bandwidth for harmonic injections: up to 25th or 50th or even 100th harmonics injection for stability tests.

– Extension of use for component and sub-system tests. Also, mobility of equipment to test installed DUTs in the field.

The three main parts of the generic converter-based test rig, which is shown in Figure 3, are introduced as follows.

## 3.1 Device Under Test

Device Under Test (DUT) can be one or more numbers of a whole WT or its sub-systems such as a nacelle consisting of converters and generator, or only converters of a WT. Nowadays, WTs are mainly full-converter or DFIG types in new developed WPPs. The main objective of test facilities is to perform compliance electrical and mechanical tests in the WT and sub-system test levels on DUT. Test results are used to evaluate the behavior of DUT during dynamic and steady-state operations according to the test standards.

## 3.2 Grid Emulator

The grid emulator or CGI consists of two back-to-back converter units to emulate a real grid characteristics for DUT, as it is shown in Figure 3. The first converter unit is connected to the utility grid through a transformer, which is called as "Active Rectifier Unit (ARU)". Generally, the control objective for the ARU is to regulate the DC-link voltage in a reference value within an acceptable deviation range. The reference value for DC-link depends on the type and objectives of the test. Thus, the ARU should perform as a current source to exchange active and reactive power between the DC-link capacitors and the utility grid.

The second converter unit is connected to the DUT through a transformer, which is called as "AC Grid Emulator". The controller of the AC grid emulator is designed to emulate a realistic grid dynamic and steady-state behavior. Besides, to have an acceptable range of total harmonic distortion and to prevent unwanted harmonics and noise interference in the setup, appropriate passive filters on both sides of the converters have been considered. Also, in some cases, active filtering methods are implemented by additional control strategies such as selective harmonic elimination and interleaved harmonic elimination methods, to decrease the need for the large passive filters (Gevorgian et al. (2016) and Averous et al. (2017)). High power and short circuit capacity are achieved by parallel connection of converters in each converter unit as indicated by N(ARU) and M(AGE) in Table 1. Thus, by this structure, the power flow in the CGI is controlled. Meanwhile, the assessment of DUT behavior would be accomplished by online simulations, measurements, and data analysis.

## 3.3 Wind Torque Emulator

Assessment of electro-mechanical interactions of WTs can be achieved by using the wind torque emulator part in the test bench. As it is shown in Figure 3, the wind torque emulator would be connected directly to the DC-link of CGI as a common DC-link, or have a separate converter unit connected to the utility grid. Separate DC-link for the wind torque emulator enables an independent control system and reduces the side-effects of power electronic converters on each other such as harmonics interference, DC-link voltage deviations, and control interactions.

The wind torque emulator is a prime mover system consisting of a drive converter connected to an AC or DC motor. This way, the characteristics of the missing WT rotor in the laboratory environment would be recreated. This objective is necessary for hardware-in-the-loop (HiL) testing of DUTs, especially for the tests, such as LVRT capability test, in which a realistic emulation of rotor torque for the DUT's main shaft is required. This requirement implies an accurate emulation of steady-state and dynamic torque characteristics of the rotor including the rotor inertia and its eigen-frequencies as studied in Neshati (2016). The drive system converts the electrical power to the mechanical power for the shaft of the generators. On the other hand, the generators convert the mechanical power to the electrical power in connection to the CGI. In this way, the power flow circulates through the utility grid, wind torque emulator, and grid emulator. The first constraint of this power circulation is the manageable power loss. Also, the second constraint for the power flow is during the LVRT capability test. During voltage sag emulation by the AC grid emulator for the DUT, the ARU has to provide the active power to the wind power emulator. Thus, the maximum required power flow and power losses during tests should be considered in the cooling system design.

## 4    Test Bench Characteristics

The advanced specification of converter-based test equipment not only makes it possible to perform grid connection compliance tests, but also gives the opportunity to analyze, understand, and predict possible challenges facing wind energy technology, and even further to develop solutions and perform validation tests. In this section, electrical characteristics of the emulated grid by an advanced test bench have been studied.

### 4.1    Emulated Grid Characteristics

The characteristics of a real power system that test article is exposed to at its Point of Common Coupling (PCC) can be emulated by CGI coupled with RTDS through high-bandwidth Power-Hardware-in-the-Loop (PHiL) interface. This way, the grid emulator can replicate all characteristics of PCC for testing DUT. The AC grid emulation can provide flexible options regarding the electrical characteristics of power grids, including impedance, short circuit ratio, inertia, and background noise.

#### 4.1.1    Grid Impedance

One of the main differences between AC and HVDC connections is the structure of equivalent grid impedance as shown in Figure 2. Especially in AC-connected offshore WPPs with long AC export submarine cables, the grid impedance is high and frequency-dependent, which can create resonances and instability (Kocewiak et al., 2013). Also, in the case of onshore AC connections, the main issue would be considerably high grid impedance for remote WPPs. Typically, for AC offshore connections, the grid impedance would be considered capacitive, while for AC onshore connections, it would be high inductive

impedance. Besides, regarding HVDC-connected offshore WPPs, the equivalent resistance of the grid impedance is low. Thus, the natural resonance damping capability in such grids is low, and the converters of WTs are prone to interact with the converters of the HVDC system. Therefore, the harmonic stability of an HVDC connection is very vulnerable. The interactions among grid impedance, converters' controllers, and passive filters can cause instability and resonance problems in a WPP as well as HVDC station (Buchhagen et al. (2015); Kocewiak et al. (2013); Sowa et al. (2019) and Beza and Bongiorno (2019)).

In a synchronous-generator-based grid, large electrical loads facilitate the grid stability during dynamics and resonances. However, in such grids, the sub-synchronous control interactions between WTs and series compensated transmission lines, which is investigated in Chernet and et al. (2019), are still a serious concern. The impedance of the test bench would be arranged as such to study the sub-synchronous control interaction as well. Therefore, it is essential to consider the emulation of grid impedance characteristics in the test environment and test results. The controllable dynamic impedance emulation is

another advantage of the converter-based CGI, in comparison to the voltage divider test equipment shown in Figure 1, which imposes fewer uncertainties regarding equivalent impedance to the point of connection of DUT.

#### 4.1.2    Short Circuit Ratio

As the AC system impedance increases, the voltage magnitude of the AC system becomes even more sensitive to the power variations at the PCC. This dependency is usually determined by the Short-Circuit Ratio (SCR), which is a ratio of the short-

circuit capacity ($S_{sc}$) versus the rated power of the AC grid at PCC ($P_{npcc}$) as illustrated in equations (1) and (2) (IEEE Std. 1204, 1997).

$$S_{sc} = \frac{V_{pcc}^2}{Z_{grid}} \tag{1}$$

Where $Z_{grid}$ is the equivalent impedance of the grid and $V_{pcc}$ is the nominal phase-to-phase voltage at PCC.

$$SCR = \frac{S_{sc}}{P_{npcc}} \tag{2}$$

The investigations in Fan and Miao (2018) have shown that a weak grid interconnection of an AC-connected WPP (e. g., ERCOT, USA) can lead to poorly damped or undamped voltage oscillations. The SCR evaluation for an HVDC-connected AC grid is defined as an Effective Short Circuit Ratio (ESCR). ESCR is the ratio of the short-circuit power of the AC grid along with HVDC converter filters and capacitor banks ($S_{(AC+HVDC)}$) to the rated power of the HVDC link ($P_{HVDC}$), as presented in equation (3). Typical weak HVDC-connections have ESCR less than 2.5 (Yogarathinam et al., 2017).

$$ESCR = \frac{S_{(AC+HVDC)}}{P_{HVDC}} \tag{3}$$

The HVDC transmission limitations imposed by AC system strength, AC grid impedance characteristics and converter Phase-Locked Loop (PLL) parameters have been investigated in Zhou et al. (2014). These studies have concluded that the operation of the HVDC converter is greatly affected by the angle of the AC grid impedance. As the impedance becomes more resistive, the minimum required SCR for the rectification side converter of the HVDC system increases; In contrast, it decreases at the inverting side converter. Also, the results have proven that the gains of the PLL, significantly affect the operation of the HVDC converter, particularly at low ESCRs (less than 1.3). In the case of stronger AC networks, the converters' control systems operate well as long as the PLL gain is preserved adequately large to achieve a satisfactory damping coefficient (Zhou et al., 2014).

The converter-based test bench has a similar structure to an HVDC connection system with two back-to-back converters. Thus it can be used to emulate an HVDC system with different ESCRs for DUT. These emulations would be implemented by adjusting the control system, modular selection of the CGI converters and reconfiguration of output filter components, especially in a test setup consisting of an RTDS system.

### 4.1.3 Grid Inertia

The grid inertia is another important criterion for evaluation of grid strength. The effective inertia constant ($H_{dc}$) for an HVDC-connected AC grid is defined as the ratio of the total rotational inertia of the AC system ($E_{TI}$) in MW-s to the MW rating of the HVDC link, which is illustrated in equation (4).

$$H_{dc} = \frac{E_{TI}}{P_{HVDC}} \tag{4}$$

$H_{dc}$ is less than 2.0 for weak grids (Yogarathinam et al., 2017). In an HVDC-connected offshore WPP, there is no rotating mass. Therefore the inertia is zero. The test bench converters can be considered as an HVDC system connection for DUT. In this

way, by adjusting the CGI control system, it is possible to emulate different inertia ranges to evaluate the control performance of WTs.

### 4.1.4 Background Harmonics

The background noise and harmonics are high-frequency content in the grid voltage as part of harmonic sources. By increasing converter-based installations, the harmonic injection and interactions have concerned the power system operators and WPP developers. The possible harmonic challenges can be studied in two main categories as follows:

- Harmonic emission sources: Non-ideal power sources and non-linear loads generate harmonics. The harmonic emission is a power quality issue and would be assessed by measurements data analysis (Sørensen et al., 2007). From power quality point of view, the harmonic emission is important because of power loss, system operation interference, and effect on overall cost. The assessment of emission limits for the connection of distorting installations at medium and higher voltage levels is recommended in IEC 61000-3-6 technical report. The emission limits depend upon the consented power of the connected power plant and the system characteristics (Joseph et al., 2012).

- Harmonic stability issues: Primarily, harmonic stability problems are significant in the case of fully renewable-based power grids; since converters mostly dominate such grids. Therefore, HVDC-connected offshore WPPs are the main subject of harmonics and resonance studies. As an example, BorWin1, which is the first offshore HVDC station and is developed to transmit wind energy from BARD offshore WPP to the onshore grid in Germany (Buchhagen et al., 2015). So far several serious problems such as outages of the HVDC station, severe harmonic distortion, and resonances in the offshore grids, have been reported because of harmonic interactions among active components such as power converters, and passive components such as filters and grid impedance (Buchhagen et al. (2015); Kocewiak et al. (2013); Bradt et al. (2011)). Besides, it is crucial to consider that the current limit recommendations in the standards do not apply to harmonic currents that are absorbed by the WPPs from the background harmonic source of the grid. Therefore, series and parallel resonances from the capacitive collector cable can easily occur in the WPP, by absorbing more harmonic current than determined in the standards (Kocewiak et al. (2017); Preciado et al. (2015)). One of the promising study proposals for the harmonic stability of converter-based power systems is impedance-based analysis (Sun, 2011).

According to Bradt et al. (2011), in the case of harmonic studies, the utility grid is characterized by two groups of parameters: The first category is the background voltage distortion present at the PCC without connection of the WPP. The second category is the driving-point impedance of the grid at harmonic frequencies, which consists of transmission system harmonic impedance and reactive compensation equipment equivalent impedance. The harmonic content of the synchronous generator-based grids would contain low order harmonics due to non-linear loads. While a converter-based grid mainly would have high order harmonics generated by high-frequency switching concepts of the power converters. Therefore, it is essential to emulate more realistic grid background harmonics using test equipment and evaluate the performance of DUT with the presence of the grid harmonics. However, high order harmonic injection would need high bandwidth in the output transformer of the AC grid emulator and the measurement instruments.

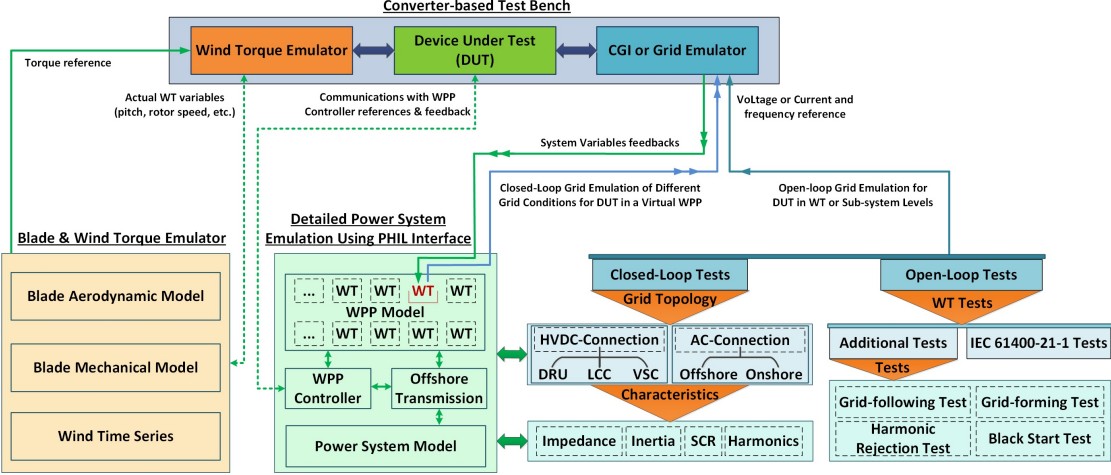

**Figure 4.** Proposed test structure for converter-based test benches.

## 4.2 Utility Grid Effects on a Test Bench

The interconnection of the grid emulating CGI and the utility grid depends on their characteristics. If the utility grid had low
SCR, then the CGI connection to the utility grid would be very similar to an HVDC connection to a weak AC grid. According
to Durrant et al. (2003), using current vector control for converters, only 0.4 per-unit (pu) power transmission can be obtained
for DC-link, where only in one of AC sides of the CGI (DUT or utility grid sides), the SCR is 1 pu. However, by using more
efficient control methods or increasing DC-link capacitance, it can be increased to higher than 0.8 pu (Zhang and Harnefors,
2011). Also, the connection of CGI to the utility grid should comply with the local grid connection requirements regarding
power quality aspects. Therefore, it is vital to consider the local grid characteristics and connection requirements in design and
control strategies for the test bench.

## 5 Proposed Test Options for Advanced Test Benches

In general, control of a WPP is managed by two control levels: WPP control level, and WT control level. The control system
of a controllable grid interface (CGI) can be designed considering objectives of tests, specification of DUT and research and
development studies. The test of DUT should be performed as such to ensure that the emulators would not affect the test
results. In Figure 4, the proposed test structure for advanced test benches is illustrated. Depending on the test modes and study
objectives, the reference values for the controllers of the test bench converters would be prepared using either Power-Hardware-
in-the-Loop (PHiL) interface or real-time system model calculations (Koralewicz et al. (2017); Averous et al. (2017)). The
electrical test options consist of two main groups including open-loop and closed-loop tests. The open-loop tests include the
capability evaluation tests in WT and sub-system levels and recreate the grid events according to predefined references and
waveforms for the CGI converters. The open-loop tests consist of IEC 61400-21-1 standard compliance tests and additional

proposed tests including grid-following, grid-forming, blackstart, and harmonic rejection capability tests. The second group of tests are proposed for validation of grid interactions in a system level including different grid connection topologies and characteristics. The closed-loop tests would analyze the behavior of DUT in connection to a virtual WPP by online simulation of a detailed power system.

Besides, the blade and wind torque control unit for wind torque emulator would be necessary in the case of WT's nacelle tests. Typically, the nacelle of WT contains gearbox, generator, converters, and output transformer. Since the mechanical parameters vary slower than electrical parameters, the speed or torque references can be defined as set-points in short-term studies for electrical tests. However, for long-term studies, the aerodynamics, pitch control, and mechanical torque could be considered in control of the wind torque emulator (Neshati, 2016). According to Figure 4, the torque or speed references for the drive system can be derived from real-time calculations based on blade aerodynamics and mechanical models, and wind speed time series. The control methods for converter-based CGI have been discussed in Gevorgian et al. (2016),Zeni et al. (2016),Espinoza et al. (2019), Espinoza et al. (2015) and Neshati (2016). In the following sections the IEC 61400-21-1 standard tests and additional proposed open-loop tests for WT capabilities as well as the proposed closed-loop tests are introduced.

## 5.1 IEC 61400-21-1 Standard Open-Loop Tests

Nowadays, most of the industrial test benches have been focused on performing the grid connection compliance tests, which are recommended in IEC 61400-21 standard. Therefore, in this section, the electrical characteristics to be simulated and validated for wind turbines are studied according to the IEC-61400-21-1 standard (IEC 61400-21-1, 2019).

### 5.1.1 Power quality aspects

The power quality tests consist of measurement of harmonic emissions and flicker tests. Flicker addresses the voltage fluctuations imposed by WTs under continuous and switching operation conditions. Mainly, the flicker effect is considerable for the first generation of WTs without power converters, which were widely connected to distribution power systems in the previous millennium. The harmonic emission consists of current harmonics, inter-harmonics (non-integer multiples of the fundamental frequency), and higher frequency components during continuous operation.

The power quality of the emulated AC grid can be arranged based on the emulation type, including HVDC or AC connection. Accordingly, the power quality aspects can be emulated for DUT. The flicker can be generated by adding a low-frequency component to the fundamental frequency of reference signals for the AC grid emulator unit. In addition, to study the harmonic interactions of WTs in a WPP, the harmonic injection tests have been considered in several test sites (Gevorgian et al. (2016); Sun et al. (2019)). Depending on the converter switching frequency of the AC grid emulator, output filter, and transformers' bandwidth, part of low order harmonics can be injected to the connection point of DUT. To date, there is no dedicated standard or regulation for harmonic interaction studies.

### 5.1.2  Steady-state operation test

The steady-state operation test evaluates the active power (P) production against wind speed, maximum power, and reactive power (Q) capability of DUT. These characteristics aim to validate the power-speed and P-Q curves. The test procedure and necessary measurements have been recommended in IEC 61400-21-1 (2019).

### 5.1.3  Control performance test

Active and reactive power related controls by WT can be divided into two major categories: WT level control and WPP level control. Control performance testing of each of these categories requires special technique. The methods discussed in this section are related to the WT level control. In this way, control performance refers to the ability of a WT in terms of active and reactive power control and grid frequency support. The assessment of power control performance is verified by set-point tracking speed and steady-state error of the control system. Also, the grid frequency support includes the active power reduction as a function of the grid over-frequency conditions. Providing additional active power during under frequency events is another grid frequency supporting feature, which should be evaluated through the relevant tests.

### 5.1.4  Transient performance test

The transient performance or Fault-Ride Through (FRT) capability consists of Low Voltage Ride-Through (LVRT) and High Voltage Ride-Through (HVRT) capabilities. Within the last decade, several serious WT tripping incidents have been reported in different countries such as Germany, China, and the UK due to voltage dips (under-voltage) and swells (over-voltage). Voltage transients have led to cascaded system trips, over-voltage excursion in transmission systems, and serious frequency deviations in power grids (Langstadtler et al. (2015); Wiser et al. (2018); Zhang et al. (2016)). Also, the measurements on real WPPs have shown that during HVDC converter blocking, the voltage at the WT terminals may increase by 30%, and even it can spike up to 2.0 pu by other transient processes (Erlich, 2016). These incidents have indicated the necessity of HVRT and LVRT capabilities for WTs. Consequently, by facing similar problems, some countries, such as Germany, Denmark, Spain, the USA, Italy, and Australia, have adapted the national network codes for both HVRT and LVRT capabilities. Accordingly, the FRT capability demands the WTs to tolerate a specified range of high- or low-voltage events for certain periods.

The compliance tests can be implemented by giving open-loop voltage reference values for the AC grid emulator as a voltage-time profile according to the network codes. In the case of the LVRT capability test, the Active Rectifier Unit (ARU) would decrease the DC-link voltage to achieve an efficient modulation index and less voltage stress on switches and filters of the AC grid emulator. However, this is not possible in cases that wind torque emulator is connected directly to the ARU.

So far, the solutions for the HVRT capability test using full-converters have been either utilization of step-up tap transformers or over-designing of the converters to be able to generate the required over-voltage range. In the case of converters over-design, the ARU should increase the DC-link voltage to make the over-voltage emulation possible for the AC grid emulator. However, using a step-up tap transformer, the nominal output voltage of the converters would be set as such to generate the maximum over-voltage at the output terminal of the transformer, which is connected to DUT.

One of the critical specifications of a test setup for FRT tests is the Rate of Change of Voltage (RoCoV) during the emulation of voltage dynamics for DUT. The AC side converter should be able to simulate over-voltage or under-voltage events very fast. This is one of the main advantages of converter-based CGIs that can emulate 100% voltage changes within less than 1 cycle of the fundamental frequency of the grid. The fastness of a converter depends on ESCR, DC-link capacitors, short circuit current capability of the AC grid emulator, control system, and overall system delays.

Furthermore, one of the recent studies in dynamic performance is the response of WTs against unbalanced faults. The unbalanced voltage deviations can be performed by setting positive and negative sequences in the voltage references and control loops for the AC grid emulator. The emulation of faults with zero-sequence voltage by CGI is challenging. However, this type of fault emulation is not necessary because of the transformers and three-wire structure of WTs. In such structures, the zero-sequence does not propagate to the WTs. However, the objective of tests with zero-sequence voltage would be the assessment of a four-wire sub-system with grounded wire.

### 5.1.5  Grid protection test

Grid protection tests refers to the disconnection and re-connection functions of a grid-connected WT following its different protection schemes. Protection schemes for disconnection from the grid operate during extreme amplitude changes or the rate of changes in voltage and frequency of the grid. The relevant test procedure to the protection schemes evaluation is provided in IEC 61400-21-1 (2019).

### 5.2  Additional Proposed Open-Loop Tests

In this section, the additional open-loop tests to the IEC 61400-21-1 standard regarding WT capabilities are proposed, as it is presented in Figure 4. The higher importance of the WT capability tests is because the wind turbine manufacturers are developing their products with advanced features that are required to be verified following the appropriate test standards and regulations. Therefore, it is urgent to foresee the near future needs in the standards. Besides, the grid connection compliance tests would be used for design validation of wind turbines or their subsystems as well.

### 5.2.1  Grid-following capability test

The electrical characteristics, which are considered in the IEC 61400-21-1 standard, only concern the performance of DUT in grid-following mode. In this way, the WTs are considered as current sources that follow the frequency and voltage references of the connected grid. Therefore, the grid-following capability of DUT addresses the control performance test, which is done for the nacelle of WTs in industrial test benches. However, this test is applicable in WT and WPP levels using the PHiL interface, as well.

### 5.2.2 Grid-forming capability test

Recently, the grid protection ability of WTs has been extended to a new capability, called "grid-forming capability". WTs with grid-forming capability can perform as a voltage source to form a local AC network during disconnection from the main power grid and supply local loads. Some manufacturers have designed a new generation of WTs with more flexible features such as the grid-forming capability to enhance the stability and reliability of converter-based power generation and the interconnected power systems. According to the grid connection requirements, WTs are allowed to disconnect from the AC grid during very severe voltage or frequency deviations out of their tolerable ranges. However, grid-forming WTs can support local loads and increase the reliability of WPPs (Tijdink et al., 2017).

Test bench converters can simulate fault occurrence conditions for DUT to evaluate the grid-forming capability of such WTs. During the grid-forming operation of DUT, the CGI should perform as a current source converter and active load for the DUT. This study case would be more challenging when the WTs are meant to be used in an HVDC-connected offshore WPP in which there is no considerable local load for the offshore WPP. In all cases, the grid-forming capability is a temporary operation mode, which would be followed by reconnection to the grid and resuming the normal operation.

### 5.2.3 System restoration and black start capability test

Following a partial or complete shutdown, it is crucial to restore the defected network and stabilize the overall power system. System restoration is the capability of reconnection of WTs to the grid after an incidental disconnection caused by a network disturbance. According to European network codes (Commission Regulation 631, 2016), the system restoration requirements consist of black start, island operation, and quick re-synchronization capabilities. State-of-the-art WTs can be equipped with functions such that they can start and run without the need for external auxiliary power supplies (Jain et al., 2018). Black start capability is one of the advanced features of WTs, which helps fast and environmentally friendly power system restoration. The black start would be essential for the start-up of a power generation unit or restart after shutting down due to faults. In a WPP, after the system shuts down, part of WTs with black start capability should be energized by an internal storage system. Then, the energized WTs should be able to energize the rest of WTs by producing wind power over time (Tijdink et al. (2017) and Jain et al. (2018)). A similar process has been described for the black start of converters of an HVDC station (Commission Regulation 631, 2016). The performance of DUT during system restoration conditions can be studied using advanced converter-based test benches.

### 5.2.4 Harmonic rejection test

The harmonic stability and harmonic interactions of wind turbine control systems can be studied by injecting harmonic voltages and currents to the wind turbine terminals using the test bench converters. This way the harmonic rejection capability and durability of WTs and their control system can be evaluated. The experimental verification of the impedance-based stability analysis method for harmonic resonance phenomena is presented in Sun et al. (2019). Besides, in the modern test setups, harmonic injection ability is considered as an advantage using CGI converters or additional equipment (Averous et al. (2017);

Gevorgian (2018); Jersch (2018); Rasmussen (2015)). Accordingly, converters' response to the specifically injected harmonics would help to analyse harmonic interactions with wind turbine control system.

## 5.3 Proposed Closed-Loop Tests

In this section, the closed-loop tests are proposed concerning the grid integration challenges of WPPs, such as HVDC system interaction, weak grid conditions, sub-synchronous, and harmonic resonances. Different grid topologies and characteristics are considered in the proposed test options to emulate a more realistic grid connection for DUT. It is evident that it is not feasible to simulate all different aspects of a real power system for a WT or WPP; however it is possible to assess part of most critical conditions in a test environment and validate the simulation models (Ausin et al. (2008); Zeni et al. (2016)).

### 5.3.1 Detailed power system emulation

The IEC 61400-21-1 standard considers the tests for a single WT, or it's sub-systems, which can be performed by CGI converters. However, these tests do not address the electrical power grid interconnection issues, such as converter interactions in the WPP level, grid characteristics influences, renewable power generation integration, and power system stability issues. Detailed power system emulation can be performed through a Power-Hardware-in-the-Loop (PHiL) interface. According to Figure 4, the voltage, current, and frequency references for the CGI converters can be extracted from the overall system model, including WPP, transmission system, and power system models. The CGI would emulate the characteristics of the AC grid for DUT in its connection point to the simulation model. The detailed power system emulation using the PHiL interface would be an effective option to perform several tests on the WPP level and analyze the behavior of DUT in extensive system conditions.

### 5.3.2 SCR and inertia emulation test

SCR of the interconnected AC grid has an essential impact on the behavior of WTs. Emulation of a variable SCR and X/R ratio allows studying the control system and stability of WTs. The number of converter modules and DC-link capacitors modifies the rating power and ESCR of the AC grid emulator. Besides, the software options for variable ESCR are considering a virtual impedance and current and power limits in the control loops of converters. In Wang (2015), the virtual impedance control method for a converter has been studied in detail. Accordingly, the virtual impedance controller behaves as a series-connected impedance at the output of a voltage source converter, which can be implemented using the RTDS system.

The magnitude of feasible inertial response by WT generator and related stability implications would be highly dependent on the location of the WPP in the power grid and SCR of PCC. The grid emulator would allow exploring these limits using the RTDS system and relevant control schemes for the CGI converters. Therefore, it is possible to emulate all inertia range from the conventional generation ($H_{dc}$=14s) down to HVDC-connected offshore grids ($H_{dc}$=0s) in a test environment to assess the performance of WTs. In Zhu (2013), the inertia emulation control method using converters of an HVDC system is proposed. It is shown that the inertia of a voltage source converter depends on the number of capacitors, DC-link voltage, and output frequency. Therefore, these options can be used for inertia emulation by CGI.

### 5.3.3 Different grid connection test

As it is described in section 4, AC and HVDC transmission systems impose different electrical characteristics and control schemes on WPPs. The converter-based CGIs allow emulating these differences in a test environment for the DUT. The control and operation system of an HVDC system depends on the structure of the HVDC converters as well. Typically, there are three topologies for the HVDC converters illustrated in Figure 4-b: Line Commutated Converters (LCC-HVDC), Voltage Source Converters (VSC-HVDC), and Diode Rectifier Units (DRU-HVDC) Göksu et al. (2017). The CGI converters have IGBT or IGCT switches in reversed-parallel connection with diodes. The converter switching method can be adjusted to perform switching based on the type of emulated HVDC topology.

The DRU-HVDC system is a cost-effective option to be used in offshore wind power transmission. To replicate a DRU-HVDC, all of the test-side converter switches should be turned off, and the remaining diodes can operate as a DRU converter. On the other side, the ARU should perform DC voltage regulation. The control methods for DRU-HVDC connected offshore WPPs have been studied in Göksu et al. (2017).

## 5.4 Discussion

The test structure for converter-based test equipment is proposed and studied in two main groups, including open-loop and closed-loop tests. The demand for open-loop tests is urgent due to developments in the WT design and manufacturing process. As it is discussed through this section, the state-of-the-art test benches are adjustable to perform tests regarding the new capabilities of WTs, mainly by new control schemes for the converters of the test bench. Besides, the use of RTDS systems for online simulations and high-speed communications in the test bench would make it feasible to implement the closed-loop tests. This way, the increasing challenges regarding operation and control of WPPs can be simulated in a test environment. The proposed test structure covers the needs of industry and research and development studies regarding the compliance test of WTs and assessment of WPPs. Furthermore, some parts of the tests, such as harmonic rejection, transient performance, power quality, and control performance, would be useful for the design validation of WT and its sub-systems as well. On the other hand, both groups of tests would be helpful to validate simulation models in WPP as well as WT levels. Therefore, the tests on the DUT can be performed as such that the results to be transferable for higher levels including WT and WPP levels.

The future works would involve in implementation of the proposed additional test options and measurement data analysis. The authors aim to propose and evaluate new test methods using available advanced test benches to increase their beneficial applications and reduce the necessity of field tests, which are difficult and costly.

## 6 Conclusions

In this paper, the generic topology of industrial test benches has been proposed. According to the structure of available industrial test benches, there is a strong potential for general harmonized topology and methods for test and assessment of WTs and WPPs. Primarily, the focus of IEC standard tests had been on the compliance test of WT capabilities through predefined

open-loop tests. The new features of modern WTs, such as grid-forming, system restoration, black start, harmonic rejection, and frequency support capabilities, have been introduced by manufacturers to support renewable energy dominated power grids. These new features necessitate new or reformed test standards in the near future. Therefore, the appropriate additional test options for newly developed capabilities are proposed. Besides, increasing challenges in wind energy integration, such as control interactions, harmonic resonances, and grid characteristics influences, have compromised the renewable generation based power grids. In this regard, the closed-loop test options for the grid interaction tests concerning different grid characteristics and topologies are proposed. The electrical characteristics of different grids consist of impedance, inertia, and SCR. In addition, the grid topologies include AC and HVDC transmission systems, as well as different HVDC converter types. By real-time simulation of the detailed power grid, the wind integration challenges can be emulated in WT and WPP levels.

Most of the available advanced test sites are developed based on full-converters. Therefore, the characteristics of a real power system can be emulated by the grid emulator coupled with RTDS systems through a high-bandwidth PHiL interface. Although it is not feasible to simulate all different aspects of a real power system; however, it is possible to assess part of the most critical conditions in a test environment and validate the simulation models for WTs and WPPs. Besides, part of the proposed tests would be applicable for the design validation of WTs or their subsystems. This way, the possibility of research, development, and demonstration studies on wind turbines and wind power plants would increase.

*Author contributions.* BN developed the new ideas of the paper with intense supervision of PES. BN and PES contributed to writing the original draft. VG and ÖG did a helpful review and editing of the paper. BN, PES, and VG developed the general test bench structure. All co-authors contributed to the control methodology and plotting of figures. BN prepared the manuscript with contributions and revisions from all co-authors.

*Competing interests.* No competing interests are present.

*Acknowledgements.* This work has received funding from PROMOTioN project as part of the European Union's Horizon 2020 research and innovation program under grant agreement No. 691714.

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
