# Peer review of "Generic Characterization of Electrical Test Benches for AC- and HVDC-Connected Wind Power Plants"

_Wind Energy Science, 2019_

## Referee Comment (RC1) · Ola Carlson (Referee) · 3 Jan 2020

Dear Authors, The paper gives a good overview of the standards and proposals for standards of tomorrow and the work going on in the area. It also includes suggestions, on a high level of details, of all possible types of useful tests to ensure stable operation of a power system with a large amount of wind power in the future. Thereby, the paper gives several good proposals of future research that needs to be done in the research and industry societies. But the details are left out for future research. May I also suggest that the author make a suggestion of which work is most important for the future operation of the power system and why.

The references are well covering the subject, the author may consider including the work by Mebtu Beza and Massimo Bongiorno, "Identification of resonance interactions in offshore-wind farms connected to the main grid by MMC-based HVDC system" International Journal of Electrical Power and Energy Systems, p. 101-113 https://doi.org/10.1016/j.ijepes.2019.04.004, in line 230 of the paper. And paper: Selam Chernet, Mebtu Bihonegn Beza, Massimo Bongiorno, "Investigation of subsynchronous control interaction in DFIG-based wind farms connected to a series compensated transmission line" International Journal of Electrical Power and Energy Systems, p. 765-774 https://doi.org/10.1016/j.ijepes.2018.09.005, in line 255 of the paper.

The abbreviations in formula 4 is not clearly written in the text, see line 261-2. In line 487 PV should be PEV.

---

## Referee Comment (RC2) · BJÖRN ANDRESEN (Referee) · 9 Jan 2020

Dear Authors, Thanks a lot for the article; it gives a very good overview of existing standards and test proposals as well as future needs and possibilities for test & validation procedures, to ensure a stable grid operation. In general, I have the following proposals to improve the article:

1. Add a little bit more explanation on "Why" you mean the proposed new tests are necessary from the system perspective - system impact. (Black start, Grid forming, etc. )

2. If would be nice to elaborate a little bit more on the transferability and assessment procedures from the proposed test bench results to Wind power plant operation, e.g. by validation of simulation models. (As the title suggest to validate the characteristics of Wind power plants), as well as the limits for the test bench tests – can anything be validated on the converter based test bench.

3. It would be beneficial for the reader and understanding of the article if you could distinguish between tests, which are necessary for the design of the Wind turbine and components (Design validation) as well as tests necessary for the grid connection and interaction with the grid. E.g. will the test of harmonic background (chapter 4.1.4) / harmonic injection be relevant to validate the design of the components (design validation), as well as potential tests for new features / harmonic filtering and last but not least harmonic stability analysis. You should consider maybe to separate the tests into – design validation of the wind turbines and components, and test necessary for the grid operation under various grid conditions.

Some more specific comments:

Chapter 5.1.5 better use the wording: "Grid Protection test" not disconnection test. Figure 3. – Use bigger symbols for the Drive motor / generator Figure 2 - Add description of the filter.

Overall a very good article and only minor comments from my side.

BR. Björn

---

## Referee Comment (RC3) · Torben Jersch (Referee) · 17 Jan 2020

Dear authors, The manuscript is providing a good overview about grid integration testing on test benches and is linking to actual discussion of harmonization. It describes all relevant parts of test benches, standardized testing und future testing.

1. For me there is missing a discussion about the dynamic change of impedances, this occurs often by changing the grids topologies and with special regard during UVRT-Testing with standard inductive voltage dividers. 2. According to the wind torque emulator chapter: The intended use is emulating the wind turbine behavior combined with HiL simulations of the entire wind turbine, as mentioned it can be done either in torque

controlled or speed controlled mode. "The motor drive system is used to simulate wind profiles to the shaft of ET's generator" –this is inaccurate. Further information: Neshati, Mohsen, et al. "Hardware-in-the-loop drive train control for realistic emulation of rotor torque in a full-scale wind turbine nacelle test rig." 2016 European Control Conference (ECC). IEEE, 2016.

Specific Comments: Figure 3: Naming the grid connected converters as DC Grid Emulator is very unusual, Active front end (AFE) or active rectifier unit (ARU) would be more common Line 151: please correct to Fraunhofer IWES, Fraunhofer Institute for Wind Energy Systems Table 1: IWES CGI rating 15 MVA , Wind emulator rating 10 MW - Wind Emulator rating seems to be the motoring power, therefore the unit should be MW.

Line 202-203: the drive system is not capable of providing mechanical loads.

I agree with Björn Andresen, it's an overall good article.

Best regards Torben

---

## Author Comment (AC1) · 20 Jan 2020

Comments reply by authors: We are pleased to get your precious comments and elaborate on our paper by considering them. We are thankful for your considerations and time regarding the paper. The following revision has been done according to the comments:

1. Referee's comment: "May I also suggest that the author make a suggestion of which work is most important for the future operation of the power system and why." Authors revision: This Comment is very close to the first comment from BJÖRN ANDRESEN as well which is "1. Add a little bit more explanation on "Why" you mean the proposed

new tests are necessary from the system perspective - system impact. (Black start, Grid forming, etc. )". Regarding the referees' comments, we have revised a paragraph in the Introduction and a paragraph in the Conclusion as follows: - The paragraph revised in the Introduction: "Primarily, power quality and transient performance during faults have been essential aspects, which needed to be tested and verified. However, by increasing trends towards 100% VRG-based grids, the VRGs are required to be developed and featured by advanced capabilities to ensure the robustness and reliability of such grids. In this way, the state-of-the-art wind turbines (WTs) are under development to be upgraded to more advanced features such as grid-forming, black start, and frequency support capabilities. These new features would necessitate test and assessment standards in the near future (Langstadtler et al.45 (2015); Asmine et al. (2017) and Gevorgian et al. (2016)). Besides, by increasing wind power installations, the requirements and appropriate test methods are required to study increasing challenges such as harmonic interactions and control performance of WPPs in connection to different types of AC and HVDC transmission systems according to (Hertem et al., 2016), (Zeni et al., 2016) and (Buchhagen et al., 2015). Thus, it is essential to adapt or define new regulations, standards, and compliance test methods to analyse the developments and issues regarding wind energy." - The paragraph added to the Conclusion: "Primarily, the focus of IEC standard tests had been on the compliance test of WT capabilities. Nowadays, the new features of modern WTs, such as Grid-forming, system restoration, black start, harmonic rejection, and frequency support capabilities, have been introduced by manufacturers to support renewable energy dominated power grids. These new features necessitate new or reformed test standards in the near future."

2. Referee's comment: "The references are well covering the subject, the author may consider including the work by Mebtu Beza and Massimo Bongiorno, "Identification of resonance interactions in offshore-wind farms connected to the main grid by MMC-based HVDC system" International Journal of Electrical Power and Energy Systems, p. 101-113, https://doi.org/10.1016/j.ijepes.2019.04.004, in line 230 of the paper." Authors

revision: Done

3. Referee's comment: "And paper: Selam Chernet, Mebtu Bihonegn Beza, Massimo Bongiorno, "Investigation of subsynchronous control interaction in DFIG-based wind farms connected to a series compensated transmission line" International Journal of Electrical Power and Energy Systems, p. 765-774 https://doi.org/10.1016/j.ijepes.2018.09.005, in line 255 of the paper." Authors revision: This paper is a very good example regarding the influences of the grid characteristics on wind turbines. We decided to review and use this paper in the subsection "4.1.1 Grid Impedance" by adding new sentences as follows: "In a synchronous generator-based grid, large electrical loads help the grid stability during dynamics and resonances. However, even in such grids, the sub-synchronous control interactions between WTs and series compensated transmission lines, which is investigated in Chernet and et al. (2019), are still a serious concern. The impedance of the test bench would be arranged as such to study the sub-synchronous control interaction as well."

4. Referee's comment: "The abbreviations in formula 4 is not clearly written in the text, see line 261-2." Authors revision: We changed the abbreviation from "TSI" to "ETI" which makes more sense for the total rotational inertia of the system in MW-s, which is the energy metrics.

5. Referee's comment: "In line 487 PV should be PEV." Authors revision: Done

Finally, we would like to appreciate the precious comments from the referee again. We hope to succeed in understanding the comments and revising the paper at a satisfactory level.

Please also note the supplement to this comment:
https://www.wind-energ-sci-discuss.net/wes-2019-90/wes-2019-90-AC1-supplement.pdf

---

## Author Comment (AC2) · 20 Jan 2020

Comments reply by authors: We are delighted to receive your precious comments and elaborate on our paper by considering them. We are thankful for your precise considerations and time regarding our paper. The following revision has been prepared according to the comments:

1. Referee's comment: "Add a little bit more explanation on "Why" you mean the proposed new tests are necessary from the system perspective - system impact. (Black start, Grid forming, etc.)" Authors revision: As it was mentioned before, this comment is close to the first comment of Prof. Ola Carlson. The reason for the new tests from the

system perspective would be new features and new challenges regarding renewable energy dominated power grids. The renewables should be developed robust and reliable to achieve a 100% green power system. New features of wind turbines are meant to facilitate these goals. Accordingly, a paragraph in the Introduction and a paragraph in the Conclusion are revised as follows: - The paragraph revised in the Introduction: "Primarily, power quality and transient performance during faults have been essential aspects, which needed to be tested and verified. However, by increasing trends towards 100% VRG-based grids, the VRGs are required to be developed and featured by advanced capabilities to ensure the robustness and reliability of such grids. In this way, the state-of-the-art wind turbines (WTs) are under development to be upgraded to more advanced features such as grid-forming, black start, and frequency support capabilities. These new features would necessitate test and assessment standards in the near future (Langstadtler et al.45 (2015); Asmine et al. (2017) and Gevorgian et al. (2016)). Besides, by increasing wind power installations, the requirements and appropriate test methods are required to study increasing challenges such as harmonic interactions and control performance of WPPs in connection to different types of AC and HVDC transmission systems according to (Hertem et al., 2016), (Zeni et al., 2016) and (Buchhagen et al., 2015). Thus, it is essential to adapt or define new regulations, standards, and compliance test methods to analyse the developments and issues regarding wind energy."

- The paragraph added to the Conclusion: "Primarily, the focus of IEC standard tests had been on the compliance test of WT capabilities. Nowadays, the new features of modern WTs, such as Grid-forming, system restoration, black start, harmonic rejection, and frequency support capabilities, have been introduced by manufacturers to support renewable energy dominated power grids. These new features necessitate new or reformed test standards in the near future."

2. Referee's comment: "It would be nice to elaborate a little bit more on the transferability and assessment procedures from the proposed test bench results to Wind power

plant operation, e.g. by validation of simulation models. (As the title suggest to validate the characteristics of Wind power plants), as well as the limits for the test bench tests – can anything be validated on the converter based test bench." Authors response: This is a very good idea to mention to the transferability and model validation applications as well as the limits of converter based test benches. Therefore, maybe it would be useful to mention that the validation of simulation models can be performed for WT models as well as WPP aggregated models. The revision will be done on the final version after the discussions.

3. Referee's comment: "It would be beneficial for the reader and understanding of the article if you could distinguish between tests, which are necessary for the design of the Wind turbine and components (Design validation) as well as tests necessary for the grid connection and interaction with the grid. E.g. will the test of harmonic background (chapter 4.1.4) / harmonic injection be relevant to validate the design of the components (design validation), as well as potential tests for new features / harmonic filtering and last but not least harmonic stability analysis. You should consider maybe to separate the tests into – design validation of the wind turbines and components, and test necessary for the grid operation under various grid conditions." Authors response: According to this comment, we restructured the test proposals into two divisions: First, WT capability tests which include the available and futuristic capabilities that are developed for WT. Second group, grid interaction tests which include electrical characteristics of the grid that have influences on WT operation. The grid interaction tests would be performed by PHIL interface. The attached figure 1 illustrates the new structure:

Also, this commend raises an interesting discussion that what is the most purpose of the test standards? According to the IEC 61400-21-1 and 21-4 standards. The grid connection compliance is the main scope of these standards, which the main concern of TSOs and WPP developers. However, some of the tests would be applicable for design validation as well, which concerns WT and component manufacturers. Therefore, we would suggest mentioning that some of the tests such as harmonic injection, grid protection, control performance, and transient performance can be used for design validation of WT, components and control systems.

4. Referee's comment: "Some more specific comments: Chapter 5.1.5 better use the wording: "Grid Protection test" not disconnection test. Figure 3. – Use bigger symbols for the Drive motor / generator Figure 2 - Add description of the filter." Authors response: Done

Finally, we would like to appreciate your precious and helpful comments on the paper. We are sure that considering these comments would add value to the paper and make it more useful.

Please also note the supplement to this comment:
https://www.wind-energ-sci-discuss.net/wes-2019-90/wes-2019-90-AC2-supplement.pdf

[Figure]

**Fig. 1.** Revised test proposal structure

---

## Author Comment (AC3) · 20 Jan 2020

Comments reply by authors: We are delighted to receive your valuable comments, and elaborate and correct our paper by considering them. We are thankful for your precise considerations and time regarding our paper. The following revision has been prepared according to the comments:

1. Referee's comment: "1. For me there is missing a discussion about the dynamic change of impedances, this occurs often by changing the grids topologies and with special regard during UVRT testing with standard inductive voltage dividers." Authors response: We would add this comment to the section "4.1.1 Grid Impedance" as an

affecting factor in WT dynamic response. Also, it would be valuable to mention this issue for voltage divider-based test facility in the Introduction.

2. Referee's comment: "2. According to the wind torque emulator chapter: The intended use is emulating the wind turbine behavior combined with HiL simulations of the entire wind turbine, as mentioned it can be done either in torque controlled or speed controlled mode. "The motor drive system is used to simulate wind profiles to the shaft of ET's generator" –this is inaccurate. Further information: Neshati, Mohsen, et al. "Hardware-in-the-loop drive train control for realistic emulation of rotor torque in a full-scale wind turbine nacelle test rig." 2016 European Control Conference (ECC). IEEE, 2016." Authors response: Thanks for your precise consideration of the content. We reviewed the suggested reference and use of it in our paper as a reference and correcting the content accordingly, would improve our grasp on the wind torque part of the test bench. The changes will be illustrated in the final version.

3. Referee's comment: "Specific Comments: Figure 3: Naming the grid connected converters as DC Grid Emulator is very unusual, Active front end (AFE) or active rectifier unit (ARU) would be more common" Authors response: The idea of using the word "DC grid emulator" was because there is a potential in the converter-based test benches to be controlled as an HVDC converter. This opportunity would make it possible to perform different grid topology tests. In an HVDC system, the utility grid-connected converter is responsible for "DC grid regulation". However, still, we can change the name to "active rectifier unit (ARU)".

4. Referee's comment: "Line 151: please correct to Fraunhofer IWES, Fraunhofer Institute for Wind Energy Systems Table 1: IWES CGI rating 15 MVA, Wind emulator rating 10 MW - Wind Emulator rating seems to be the motoring power, therefore the unit should be MW. " Authors response: Done

5. Referee's comment: "Line 202-203: the drive system is not capable of providing mechanical loads." Authors response: Our perception from "mechanical load" was the

same as "mechanical torque". We would remove the incorrect part. Thanks for the comment.

Finally, we would like to appreciate your considerations and detailed comments. We hope to revise the paper as such to include all of your valuable comments.

Please also note the supplement to this comment:
https://www.wind-energ-sci-discuss.net/wes-2019-90/wes-2019-90-AC3-supplement.pdf

---

## Editor Comment (EC1) · HANNELE HOLTTINEN (Editor) · 27 Jan 2020

Dear authors, I encourage a resubmission. Based on your replies for the reviewer comments, most of them you have already considered. For the comment on what terminology to use for HiL test, I encourage to consider the terms proposed by Torben, however, it is a good idea to bring the discussion you have now in your response in the article, as it will further explain.
* * *

---

## Author Response (AR1)

(Comments received and published: 3 January 2020)

**Comments reply by authors:**

We are pleased to get your precious comments and elaborate on our paper by considering them. We appreciate your considerations and time regarding the paper. The following revision have been prepared according to the comments:

**1. Referee's comment:** "May I also suggest that the author make a suggestion of which work is most important for the future operation of the power system and why."

**Author's response:**

This Comment is very close to the first comment from BJÖRN ANDRESEN as well that is "1. Add a little bit more explanation on "Why" you mean the proposed new tests are necessary from the system perspective - system impact. (Black start, Grid forming, etc. )". Regarding to the referees comments, we have revised paragraphs in the Introduction, section 5.2 and Conclusion as follows:

**Author's changes:**

-   The paragraph revised in the Introduction, lines 41-49:

"Primarily, power quality and transient performance during faults have been essential aspects, which needed to be tested and verified. However, by increasing trends towards 100% VRG-based grids, the VRGs are required to be developed and featured by advanced capabilities to ensure the robustness and reliability of such grids. In this way, the state-of-the-art Wind Turbines (WTs) are under development to be upgraded to more advanced features such as grid-forming, black start, and frequency support capabilities. These new features would necessitate test and assessment standards in the near future (Langstadtler et al. (2015); Asmine et al. (2017); Gevorgian et al. (2016)). Besides, by increasing wind power installations, the requirements and appropriate test methods are required to study the rising challenges such as harmonic

resonances and control interactions of WPPs in connection to different types of AC and HVDC transmission systems according to Hertem et al. (2016), Zeni et al. (2016) and Buchhagen et al. (2015)."

- The paragraph added to the section 5.2  Additional Proposed Open-Loop Tests, lines 427-429:

"In this section, the additional open-loop tests to the IEC 61400-21-1 standard regarding WT capabilities are proposed, as it is presented in Figure 4. The higher importance of the WT capability tests is because the wind turbine manufacturers are developing their products with advanced features that are required to be verified following the appropriate test standards and regulations. Therefore, it is urgent to foresee the near future needs in the standards."

- The paragraph added to the Conclusion, lines 530-533:

"Primarily, the focus of IEC standard tests had been on the compliance test of WT capabilities through predefined open-loop tests. The new features of modern WTs, such as grid-forming, system restoration, black start, harmonic rejection, and frequency support capabilities, have been introduced by manufacturers to support renewable energy dominated power grids. These new features necessitate new or reformed test standards in the near future."

**2. Referee's comment:** "The references are well covering the subject, the author may consider including the work by Mebtu Beza and Massimo Bongiorno, "Identification of resonance interactions in offshore-wind farms connected to the main grid by MMC-based HVDC system" International Journal of Electrical Power and Energy Systems, p. 101-113, https://doi.org/10.1016/j.ijepes.2019.04.004, in line 230 of the paper."

**Author's response:**

Thanks for the recommended useful reference.

**Authors revision:**

The paper has been referred in line 254 and is added to the references as 43th reference.

**3. Referee's comment:** "And paper: Selam Chernet, Mebtu Bihonegn Beza, Massimo Bongiorno, "Investigation of subsynchronous control interaction in DFIG-based wind farms connected to a series compensated transmission line" International Journal of Electrical Power and Energy Systems, p. 765-774 https://doi.org/10.1016/j.ijepes.2018.09.005, in line 255 of the paper."

**Author's response:**

Thanks for the recommended useful reference. This paper is a very good example regarding the influences of the grid characteristics on wind turbines. We decided to review and use this paper in the sub-section "4.1.1 Grid Impedance"

**Authors revision:**

In the sub-section "4.1.1 Grid Impedance", the new paragraph has been added along with the suggested paper as 44th reference, lines 255-258 as follows:

"In a synchronous-generator-based grid, large electrical loads facilitate the grid stability during dynamics and resonances. However, in such grids, the sub-synchronous control interactions between WTs and series compensated transmission lines, which is investigated in Chernet and et al. (2019), are still a serious concern. The impedance of the test bench would be arranged as such to study the sub-synchronous control interaction as well."

**4. Referee's comment:** "The abbreviations in formula 4 is not clearly written in the text, see line 261-2."

**Author's response:**

We appreciate your precise attention to the paper.

**Authors revision:**

We changed the abbreviation from "TSI" to "$E_{TI}$" which makes more sense for the total rotational inertia of the system in MW-s, that is the energy metrics.

**5. Referee's comment:** "In line 487 PV should be PEV."

**Author's response:**

We appreciate your helpful comments. We corrected the typo.

Finally, we would like to appreciate the precious comments from the referee again. We hope to succeed in understanding the comments and revising the paper on a satisfactory level.

**Responses to the interactive comment on** "Proposal for Generic Characterization of Electrical Test Benches for AC- and HVDC-Connected Wind Power Plants" by Behnam Nouri et al.

**Responses to the referee: Prof. Björn Andresen**

bjra@ase.au.dk

(Comments received and published: 9 January 2020)

**Comments reply by authors:**

We are delighted to receive your precious comments and elaborate on our paper by considering them. We are thankful for your precise considerations and time regarding our paper. The following revision has been done according to the comments:

**1. Referee's comment:** "Add a little bit more explanation on "Why" you mean the proposed new tests are necessary from the system perspective - system impact. (Black start, Grid forming, etc.)"

**Author's response:**

As it was mentioned before, this comment is close to the first comment of Prof. Ola Carlson. The reason for the new tests from the system perspective would be new features and new challenges regarding renewable energy dominated power grids. The renewables should be developed robust and reliable to achieve 100% green power system. New features of wind turbines are meant to facilitate these goals. Accordingly, Abstract, Introduction and Conclusion are revised as follows:

**Author's changes:**

- A sentence has been added to the Abstract to convey the importance of WT capabilities from the system perspective, lines 4-6:

"Besides, the modern wind turbines have been featured by new capabilities, such as grid-forming, black start, harmonic rejection and frequency support, **to increase the robustness and reliability of renewable-energy-based grids.** Furthermore, the increasing challenges, such as

harmonic resonances and grid interactions, **are compromising wind energy integration into power systems.**"

- The paragraph revised in the Introduction, lines 41-50: (VRG: Variable Renewable Generation)

"Primarily, power quality and transient performance during faults have been essential aspects, which needed to be tested and verified. However, by increasing trends towards 100% VRG-based grids, the VRGs are required to be developed and featured by advanced capabilities to ensure the robustness and reliability of such grids. In this way, the state-of-the-art Wind Turbines (WTs) are under development to be upgraded to more advanced features such as grid-forming, black start, and frequency support capabilities. These new features would necessitate test and assessment standards in the near future (Langstadtler et al. (2015); Asmine et al. (2017); Gevorgian et al. (2016)). Besides, by increasing wind power installations, the requirements and appropriate test methods are required to study the rising challenges such as harmonic resonances and control interactions of WPPs in connection to different types of AC and HVDC transmission systems according to Hertem et al. (2016), Zeni et al. (2016) and Buchhagen et al. (2015). Thus, it is essential to adapt or define new regulations, standards, and compliance test methods to analyse the developments and issues regarding wind energy."

- The paragraph added to the Conclusion, lines 531-533:

"The new features of modern WTs, such as grid-forming, system restoration, black start, harmonic rejection, and frequency support capabilities, have been introduced by manufacturers **to support renewable energy dominated power grids.** These new features necessitate new or reformed test standards in the near future."

**2. Referee's comment:** "It would be nice to elaborate a little bit more on the transferability and assessment procedures from the proposed test bench results to Wind power plant operation, e.g. by validation of simulation models. (As the title suggest to validate the characteristics of Wind power plants), as well as the limits for the test bench tests – can anything be validated on the converter based test bench."

**Authors response:**

This is a very good idea to mention to the transferability and model validation applications as well as the limits of converter based test benches. Therefore, maybe it would be useful to mention that the validation of simulation models can be performed for WT models as well as WPP aggregated models.

**Author's changes:**

- To elaborate on Simulation Model Validation, the reference 18 as "CIGRE Technical Brochures: Network modelling for harmonic studies, JWG C4/B4.38, Reference no. 766, 2019" has been added and referred in Introduction line 52.

"To date, several standards and recommendations such as IEC, IEEE, DNV GL, and CIGRE have been published for design, simulation, operation, and test of electrical aspects of WTs (IEC 61400-21-1 (2019);IEEE Std 1094-1991 (1991);DNVGL-ST-0076 (2015) and CIGRE TB766 (2019)).

- To elaborate on importance of tests and theirs application in Simulation Model Validation for WTs and WPPs, the following sentences have been added in subsection "2.1.1 Electrical test levels", lines 132-135:

"The test results concern wind farm developers and system operators in terms of WPP model validation and grid connection compliance, and WT manufacturers in terms of WT design and simulation model validations. This way, the results of tests are considered to be transferable and useful for the assessment of WTs as well as WPPs and developed simulation models (Ausin et al. (2008); Zeni et al. (2016); Koralewicz et al. (2017)).

- To mention the limits of tests and application of tests for model validation the following sentences have been added to section "5.3 Proposed Closed-Loop Tests", lines 474-476:

"It is evident that it is not feasible to simulate all different aspects of a real power system for a WT or WPP; however it is possible to assess part of most critical conditions in a test environment and validate the simulation models (Ausin et al. (2008); Zeni et al. (2016))."

- In addition, the section "5.4 discussion" has been added to the paper to discuss and summarize the propose test options, limits and application of tests regarding model validation and design validation, lines 519-523:

"The proposed test structure covers the needs of industry and research and development studies regarding the compliance test of WTs and assessment of WPPs. Furthermore, some parts of the tests, such as harmonic rejection, transient performance, power quality, and control performance, would be useful for the design validation of WT and its sub-systems as well. On the other hand, both groups of tests would be helpful to validate simulation models in WPP as well as WT levels. Therefore, the tests on the DUT can be performed as such that the results to be transferable for higher levels including WT and WPP levels."

- In this regard, the following sentences have been added to the Conclusion as well, lines 542-543:

"Although it is not feasible to simulate all different aspects of a real power system; however, it is possible to assess part of the most critical conditions in a test environment and validate the simulation models for WTs and WPPs."

**3. Referee's comment:** "It would be beneficial for the reader and understanding of the article if you could distinguish between tests, which are necessary for the design of the Wind turbine and components (Design validation) as well as tests necessary for the grid connection and interaction with the grid. E.g. will the test of harmonic background (chapter 4.1.4) / harmonic injection be relevant to validate the design of the components (design validation), as well as potential tests for new features / harmonic filtering and last but not least harmonic stability analysis. You should consider maybe to separate the tests into – design validation of the wind turbines and components, and test necessary for the grid operation under various grid conditions."

**Authors response:**

According to this comment, we restructured the test proposals into two divisions: First group is open-loop tests, which include the available and futuristic capability tests that are developed for WTs. Open-loop tests would be implemented by predefined references and waveforms for the converters of the grid emulator. Second group is closed-loop tests, which includes electrical characteristics of different grids and evaluates grid interactions with WTs and WPPs. The closed-loop tests would be performed by PHIL interface.

Also, this commend raises an interesting discussion that what is the most purpose of the test standards? According to the IEC 61400-21-1 and 21-4 standards. The grid connection compliance is the main scope of these standards, which the main concern of TSOs and WPP developers. Besides, some of the tests would be applicable for design validation as well, which concerns WT and component manufacturers. Therefore, we would suggest mentioning that parts of the tests can be used for design validation of WT, components and control systems.

**Author's changes:**

- The new structure of Figure 4 is illustrated below:

[Figure]

- The explanation of the revised test structure in the Abstract has been changed to the following sentences, lines 7-13:

"This paper proposes a generic test structure within two main groups, including open-loop and closed-loop tests. The open-loop tests include the IEC 61400-21-1 standard tests as well as the additional proposed test options for the new capabilities of wind turbines, which replicate grid connection compliance tests using open-loop references for the grid emulator. Besides, the closed-loop tests evaluate the device under test as part of a virtual wind power plant and perform real-time simulations considering the grid dynamics. The closed-loop tests concern grid connection typologies consisting of AC and HVDC, as well as different electrical characteristics, including impedance, short circuit ratio, inertia, and background harmonics."

- To elaborate on Electrical Design and Validation, the 16th reference as "IEEE Std 1094-1991: IEEE Recommended Practice for the Electrical Design and Operation of Windfarm Generating Stations, 1991" and the 17th reference as "DNVGL-ST-0076: Design of electrical installations for wind turbines, 2015" have been added to the Introduction, line 52:

"To date, several standards and recommendations such as IEC, IEEE, DNV GL, and CIGRE have been published **for design, simulation, operation, and test** of electrical aspects of WTs (IEC 61400-21-1 (2019);IEEE Std 1094-1991 (1991);DNVGL-ST-0076 (2015) and CIGRE TB766 (2019))."

- To elaborate on importance of tests for different industrial partners, the following sentences have been added in subsection "2.1.1 Electrical test levels", lines 132-135:

"The test results concern **wind farm developers and system operators** in terms of WPP model validation and grid connection compliance, and **WT manufacturers in terms of WT design and simulation model validations**. This way, the results of tests are considered to be transferable and useful for the assessment of WTs as well as WPPs and developed simulation models (Ausin et al. (2008); Zeni et al. (2016); Koralewicz et al. (2017))."

- In addition, the section "5.4 discussion" has been added to the paper to discuss and summarize the propose test options, limits and application of tests regarding model validation and design validation, lines 520-521:

"Furthermore, some parts of the tests, such as harmonic rejection, transient performance, power quality, and control performance, would be useful for the design validation of WT and its sub-systems as well."

**4. Referee's comment:** "Some more specific comments:

Chapter 5.1.5 better use the wording: "Grid Protection test" not disconnection test. Figure 3. – Use bigger symbols for the Drive motor / generator Figure 2 - Add description of the filter."

**Authors response:**

We appreciate your precise attention and considerations regarding the content of the paper.

**Author's changes:**

The above mentioned comments have been applied on the paper.  Figure 2 has been revised. Also, the figure description has been added to the section "2. Grid Connection Compliance Tests" as well, lines 68-72:

"As shown in Figure 2-a, the AC-connected offshore WPP connects to the main onshore grid through high voltage submarine cables and transformers. The shunt inductors are required to dampen the possible over-voltage phenomena caused by the capacitive effect of the AC cables. The typical structure of an HVDC-connected offshore WPP is illustrated in Figure 2-b, which consists of HVDC transmission cables, transformers, AC/DC converters, and harmonic filters of the converters."

Finally, we would like to appreciate your precious and helpful comments on the paper. We are sure that considering these comments would add value to the paper and make it more useful.

**Responses to the interactive comment on** "Proposal for Generic Characterization of Electrical Test Benches for AC- and HVDC-Connected Wind Power Plants" by Behnam Nouri et al.

**Responses to the referee: Dr. Torben Jersch**

torben.jersch@iwes.fraunhofer.de

(Comments received and published: 17 January 2020)

**Comments reply by authors:**

We are delighted to receive your valuable comments, and elaborate and correct our paper by considering them. We are thankful for your precise considerations and time regarding our paper. The following revision has been prepared according to the comments:

**1. Referee's comment:** "1. For me there is missing a discussion about the dynamic change of impedances, this occurs often by changing the grids topologies and with special regard during UVRT testing with standard inductive voltage dividers."

**Author's response:**

We would add this comment to the section "4.1.1 Grid Impedance" as an affecting factor in WT dynamic response. Also, it would be valuable to mention this issue for voltage divider-based test equipment in the Introduction.

**Author's changes:**

- To mention the disadvantage of voltage divider regarding uncontrollable impedance dynamics, the following sentence has been changed in lines 38-40:

"However, it has certain fundamental limitations, such as dependence on a stronger point of interconnections, **uncontrollable dynamic change of impedance during test**, and inability to replicate any evolving grid characteristics.."

- The following sentences have been added to the section "4.1.1 Grid Impedance" to mention the dynamic impedance issue, lines 259-261:

"The controllable dynamic impedance emulation is another advantage of the converter-based CGI, in comparison to the voltage divider test equipment shown in Figure 1, which imposes fewer uncertainties regarding equivalent impedance to the point of connection of DUT."

**2. Referee's comment:** "2. According to the wind torque emulator chapter: The intended use is emulating the wind turbine behavior combined with HiL simulations of the entire wind turbine, as mentioned it can be done either in torque controlled or speed controlled mode. "The motor drive system is used to simulate wind profiles to the shaft of ET's generator" –this is inaccurate. Further information: Neshati, Mohsen, et al. "Hardware-in-the-loop drive train control for realistic emulation of rotor torque in a full-scale wind turbine nacelle test rig." 2016 European Control Conference (ECC). IEEE, 2016."

**Authors response:**

Thanks for your precise consideration of the content. We reviewed the suggested reference and use of it in our paper as a reference and correcting the content accordingly, would improve our grasp on the wind torque part of the test bench.

**Author's changes:**

- The recommended paper has been added to this paper as 40th reference.
- The following sentences have been added to the section "3.3 Wind Torque Emulator", lines 223-227 as follows:

"The wind torque emulator is a prime mover system consisting of a drive converter connected to an AC or DC motor. This way, the characteristics of the missing WT rotor in the laboratory environment would be recreated. This objective is necessary for hardware-in-the-loop (HiL) testing of DUTs, especially for the tests, such as LVRT capability test, in which a realistic emulation of rotor torque for the DUT's main shaft is required. This requirement implies an accurate emulation of steady-state and dynamic torque characteristics of the rotor including the rotor inertia and its eigen-frequencies as studied in Neshati (2016)."

**3. Referee's comment:** "Specific Comments: Figure 3: Naming the grid connected converters as DC Grid Emulator is very unusual, Active front end (AFE) or active rectifier unit (ARU) would be more common"

**Authors response:**

The idea of using the word "DC grid emulator" was because there is a potential in the converter-based test benches to be controlled as an HVDC converter. This opportunity would make it possible to perform different grid topology tests. In an HVDC system the utility grid connected converter is responsible for "DC grid regulation". However, still, we can change the name to "active rectifier unit (ARU)".

**Author's changes:**

The name "DC grid emulator" is changed to "Active Rectifier Unit (ARU)" in section "3.2 Grid Emulator" as well as Figure 3.

**4. Referee's comment:** "Line 151: please correct to Fraunhofer IWES, Fraunhofer Institute for Wind Energy Systems Table 1: IWES CGI rating 15 MVA, Wind emulator rating 10 MW - Wind Emulator rating seems to be the motoring power, therefore the unit should be MW."

**Authors response:**

Thanks for the precise attention. The Table 1 and name of institution "Fraunhofer IWES" have been corrected according to the comments.

**5. Referee's comment:** "Line 202-203: the drive system is not capable of providing mechanical loads."

**Authors response:**

Our perception of "mechanical load" was the same as "mechanical torque". To avoid possible miss-understandings, the phrase "mechanical load" has been removed from the section "3.3 Wind Torque Emulator". Thanks for the comment.

Finally, we would like to appreciate your considerations and detailed comments. We hope to revise the paper as such to include all of your valuable comments.

[revised manuscript text omitted]

---

## Author Response (AR2)

**Responses to the editor's comments on** "Proposal for Generic Characterization of Electrical Test Benches for AC- and HVDC-Connected Wind Power Plants" by Behnam Nouri et al.

Dr. Hannele Holttinen

hannele.holttinen@recognis.fi

(Comments received and published: 18 Feb 2020)

We are delighted to receive your precious comments and elaborate on our paper by considering them. We appreciate your decision and considerations regarding our paper. The following revision has been prepared according to the comments:

**1. Editor's comment:** "The only one unclear is the first one "a suggestion of which work is most important for the future operation of the power system and why". You state generally "These new features necessitate new or reformed test standards in the near future" - do you mean that all the new features are equally important to address already?"

**Author's response:**

In a 100% renewable based power system, renewable energy systems, such as wind power plants, should be able to maintain stability and reliability of the power system. The stability and operation of a wind power plant would depend on interoperability and capabilities of its wind turbines.

Since grid forming and black start capabilities have been already required by system operators and included in the manufacturers design considerations, these two new features would be the most important capabilities which need to be addressed in the test standards. Besides, the harmonic interactions among converters have been reported as an increasing challenge for renewable energy systems. Therefore, harmonic stability of wind power plants would be another important topic that should be studied and included in the standards. Regarding to this comments, we have revised paragraphs in the Introduction, Discussion and Conclusion as follows:

**Author's changes:**

- Introduction, lines 45-48:

"However, by increasing trends towards 100% VRG-based grids, the VRGs are required to be developed and featured by advanced capabilities to secure the robustness and reliability of such grids. The operation and stability of VRG-based power systems depend on the interoperability and capabilities of the individual power generating systems such as Wind Turbines (WT). In this way, the state-of-the-art WTs are under development towards advanced features, especially grid-forming and black start capabilities. These new capabilities necessitate appropriate test and assessment standards in the near future."

- 5.4 Discussion, lines 492-497:

"Operation and stability of WPPs depend on the interoperability and capabilities of the individual WTs. Since grid-forming and blackstart capabilities have been already required by system operators and included in the manufacturers design considerations, these two new features would be the most important capabilities which need to be addressed in the test standards. Besides, the harmonic interactions among converters have been reported as an increasing challenge for renewable energy systems. Therefore, harmonic stability of WPPs and HVDC systems would be another important topic that should be studied and included in the standards."

- Conclusion, lines 505-507:

"The new features of modern WTs, especially grid-forming, and blackstart capabilities as well as harmonic stability considerations, have been required by system operators and developed by manufacturers to support renewable energy dominated power grids."

**2. Editor's comment:** "I would also recommend to check the language of the paper to clarify the text that is in some parts having unnecessary long sentences and paragraphs."

**Author's response:**

We appreciate your considerations regarding the content quality. We reviewed the paper again concerning this valuable comment.

**Author's changes:**

We have revised the paper as such to reduce long and unnecessary paragraphs. This way, we could reduce 1 page rather than previous version as well.

**3. Editor's comment:** "Check especially the Abstract (the use of "Besides..." I think that you mean that on top of harmonizing state-of-the-art test, new features are needed) and 5.4 Discussion and Introduction (use of Part 1... instead of Section 1...?, and try to be as clear in what you are proposing to do as in Conclusions."

**Author's response:**

Thank you for the useful comments. We have considered the recommendations all over the paper.

**Author's changes:**

The word "part" is replaced with "section" all over the paper. In addition, the transparency of the paragraphs have been checked and revised.

Finally, we appreciate your kind consideration and precious recommendations. We hope to succeed in applying all of the recommendations in the new revision.

Best regards

[revised manuscript text omitted]